# Trends and disparities in dilated cardiomyopathy related mortality among adults in the United States: A CDC WONDER analysis (1999–2023)

Zian Zafar Cheema[1], Mohammad Atout[2], Taha Kassim Dohadwala[3], Ahmed Talaat Deiab[4], Aya Abouayana[5], Hazim Mesmar[6], Asmaa Hasan[7], Amaad Alam Shah[8], Muhammad Babar Mahmood[9], Daniel James Lewis[10], Hasan Ahmed[11], Maryam Shahzad[12], Mushood Ahmed[13], Nabeel Ahmed[14], Raheel Ahmed[15,16]*, Syed Khurram M. Gardezi[4]

1 Department of Medicine, Amna Inayat Medical College, Sheikhupura, Pakistan, 2 Department of Internal Medicine, Department of Medicine, The Specialty Hospital, Amman, Jordan, 3 Department of Medicine, David Tvildiani Medical University, Tbilisi, Georgia, 4 Department of Cardiology, Sheikh Shakhbout Medical City, Abu Dhabi, United Arab Emirates, 5 Department of Internal Medicine, Zayed Military Hospital, Abu Dhabi, United Arab Emirates, 6 Department of Medicine, Jordan University Hospital, Amman, Jordan, 7 Department of Medicine, Prince Hamzeh Hospital, Amman, Jordan, 8 Department of Medicine, Newcastle Medical School, Newcastle, United Kingdom, 9 Department of Trauma and Orthopaedics, Sunderland Royal Hospital, Sunderland, United Kingdom, 10 Northumbria Healthcare NHS Foundation Trust, United Kingdom, 11 Department of Medicine, Imperial College London, London, United Kingdom, 12 Department of Medicine, Dow University of Health Sciences, Karachi, Pakistan, 13 Department of Medicine, Rawalpindi Medical University, Rawalpindi, Pakistan, 14 Director Exponus Group Limited, Newcastle, United Kingdom, 15 National Heart & Lung Institute, Imperial College London, London, United Kingdom, 16 Department of Cardiology, Royal Brompton Hospital, London, United Kingdom

* R.ahmed21@imperial.ac.uk

## Abstract

### Background

Dilated cardiomyopathy (DCM) is a progressive myocardial disease characterized by ventricular dilation and impaired systolic function, often leading to heart failure and increased mortality. This study aims to analyze DCM-related mortality trends among adults in the United States from 1999 to 2023.

### Methods

Trends in DCM-related mortality among adults aged ≥ 25 years from 1999 to 2023 were analyzed using the CDC WONDER multiple-cause of death database. Age adjusted mortality rates (AAMRs) per 100,000 persons and annual percent change (APC) were calculated and stratified by year, sex, race, census region and urbanization status.

**Data availability statement:** All relevant data are within the paper and its Supporting Information files.

**Funding:** The author(s) received no specific funding for this work.

**Competing interests:** The authors have declared that no competing interests exist.

## Results

From 1999 to 2023, there were 184,073 deaths in the United States attributed to DCM. Over this period, the AAMR declined from 5.19 in 1999 to 2.34 in 2023. Between 1999 and 2002, the AAMR decreased significantly from 5.19 to 4.38 (APC −6.20* [95% CI, −12.42 to −0.72; P = 0.0208]). The trend remained stable between 2002 and 2005, followed by a significant decline between 2005 and 2014, where the AAMR decreased from 4.96 to 2.66 (APC −6.84* [95% CI, −9.00 to −3.05; P = 0.0300]). The trend then remained relatively unchanged from 2014 to 2023. In 2023, males (3.4) averaged a considerably higher AAMR than females (1.38). Among racial groups, the highest AAMR in 2023 was reported in the Non-Hispanic (NH) Black group (3.77), followed by the NH White group (2.3), the Hispanics and Latinos (1.67) and the NH Others group (NH Asians and NH Native American Indian or Alaskan Native) at 1.23. Overall from 1999 to 2020, rural areas (3.52) averaged a significantly higher AAMR than urban areas (3.50). Regionally, in 2023, the Western region averaged the highest AAMR at 2.69, followed by the South at 2.31, the Midwest at 2.42 and lastly the Northeast at 1.83.

## Conclusion

Our analysis revealed a significant decline in DCM-related mortality rates in the United States from 1999 to 2023, with the most substantial reductions occurring between 1999 and 2014. However, disparities persist, with higher mortality rates observed in males, NH Black individuals, and rural populations. Regional variations also highlight the need for targeted interventions to further reduce the burden of DCM.

## 1. Introduction

Dilated cardiomyopathy (DCM), also referred to as dilated non-ischemic cardiomyopathy, is a diverse group of myocardial disorders characterized by left ventricular (LV) or biventricular dilation and systolic dysfunction, as evidenced by a reduced left ventricular ejection fraction (LVEF). This occurs in the absence of abnormal loading conditions such as hypertension, valvular disease, or coronary artery disorders [1]. DCM is relatively prevalent, affecting approximately 1 in 250 individuals [2]. In the United States, it contributes to an estimated 10,000 deaths and 46,000 hospitalizations annually [3]. A study by Ababio et al. reported a period prevalence of DCM at 118.33 per 100,000 between 2017 and 2019 in the United States, with higher rates observed among individuals aged 65 and older, males, and African Americans [4]. Based on the 2019 U.S. population, this translates to approximately 388,350 affected individuals. After adjusting for idiopathic DCM cases, the prevalence was estimated at 59.23 per 100,000, corresponding to 194,385 individuals [4]. DCM can result from both genetic and non-genetic factors, including hypertension, valvular disease, inflammatory or infectious causes, and toxin exposure. Even in non-genetic

cases, an individual's genetic predisposition can influence disease progression, and multiple underlying mechanisms may coexist [5]. The severity of LV systolic dysfunction in DCM varies, often progressing over time [6]. While reduced systolic function does not always present with symptoms, DCM is a significant risk factor for heart failure (HF) and is frequently associated with severe arrhythmias, reflecting pathological involvement of the cardiac conduction system [7]. Despite significant advancements in HF management, DCM remains a leading cause of heart transplantation and is associated with high mortality rates [8]. Although prior research has explored DCM-related mortality, data examining its demographic and regional distribution across the United States remain limited. Previous analyses using the CDC WONDER database, such as those by Komminni et al. and Ashraf et al., assessed trends in DCM-related mortality from 1999 to 2020 [9,10]. To build upon this work, we conducted an updated analysis extending through 2023, allowing for a comparison of trends before and after the COVID-19 pandemic. Identifying these patterns is crucial for addressing ongoing public health disparities. Therefore, we performed a retrospective nationwide study using CDC WONDER to assess DCM-related mortality trends among U.S. adults from 1999 to 2023, with a focus on variations by age, sex, race/ethnicity, and geographic region in order to identify high-risk populations and potential gaps in healthcare access and outcomes.

## 2. Methods

### 2.1. Study setting and population

Deaths occurring within the United States related to dilated cardiomyopathy (DCM) were extracted from the CDC WONDER (Centers for Disease Control and Prevention Wide-Ranging ONline Data for Epidemiologic Research) database [11]. CDC-WONDER is an exhaustive repository of death certificate data from the fifty states of the USA as well as the District of Columbia. The Multiple Cause-of-Death Public Use record death certificates were studied to identify records in which DCM was mentioned as either contributing or underlying cause of death on nationwide death certificates. This database has previously been used to determine trends in mortality of DCM. Patients were identified using the International Classification of Diseases 10th Revision Clinical Modification (ICD-10-CM) codes I42.0 for DCM in individuals ≥ 25 years of age [12]. Institutional review board approval was not required for this study, as we used a publicly available, de-identified dataset provided by the government. The study adheres to the reporting standards outlined in the Strengthening the Reporting of Observational Studies in Epidemiology (STROBE) guidelines [13].

### 2.2. Data abstraction

Data on DCM-related deaths and population sizes were extracted. Demographics (sex, race/ethnicity, and age), and regional information (urban-rural and state) were extracted from 1999 to 2023. Race/ethnicities were delineated as non-Hispanic (NH) white, NH Black or African American, NH others (NH Asian or Pacific Islander, NH American Indian or Alaska Native, etc), and Hispanics or Latinos. These race/ethnicity categories have previously been used within analyses from the CDC WONDER database and rely on reported data on death certificates. For age stratification, age was divided into the following categories: 25–34, 35–44, 45–54, 55–64, 65–74, 75–84, and 85 years and older. Trends in mortality from DCM were evaluated based on state-specific variations, U.S. census regions (Northeast, Midwest, South, West), and county-level urbanization classifications. Counties were categorized as rural (micropolitan, noncore regions) or urban (large central metro, large fringe metro, medium metro, small metro) based on the 2013 National Center for Health Statistics Urban-Rural Classification Scheme [14].

### 2.3. Statistical analysis

Crude and age-adjusted mortality rates per 100,000 population were determined. Crude mortality rates (CMRs) were determined by dividing the number of DCM-related deaths by the corresponding US population of that year. Age adjusted mortality rates (AAMRs) were calculated by standardizing the DCM-related deaths to the 2000 US population as

previously described [15]. The Joinpoint Regression Program (Joinpoint V 5.1.0.0, National Cancer Institute) was used to determine trends in AAMRs and CMRs using annual percent change (APC) [16]. This method identifies significant changes in AAMRs and CMRs over time by fitting log-linear regression models where temporal variation occurred. APCs with 95% confidence intervals (CI) for the AAMRs and CMRs were calculated at the identified line segments linking join points using the Monte Carlo permutation test. APCs were considered increasing or decreasing if the slope describing the change in mortality was significantly different from zero using 2-tailed t testing. Statistical significance was set at P<.05.

## 3. Results

### 3.1. Overall

From 1999 to 2023, there were 184,073 deaths in the United States attributed to DCM. Over this period, the AAMR declined from 5.19 in 1999 to 2.34 in 2023. Between 1999 and 2002, the AAMR decreased significantly from 5.19 to 4.38 (APC −6.20% [95% CI, −12.42 to −0.72; P = 0.0208]). The trend remained stable between 2002 and 2005, followed by a significant decline between 2005 and 2014, where the AAMR decreased from 4.96 to 2.66 (APC −6.84% [95% CI, −9.00 to −3.05; P = 0.0300]). The trend then remained relatively unchanged from 2014 to 2023. (S1 Table, S2 Table, S3 Table, Fig 1).

### 3.2. DCM-related AAMR stratified by sex

Throughout the study period, males consistently had a considerably higher AAMR than females. From 1999 to 2023, the AAMR among males decreased from 7.74 in 1999 to 3.4 in 2023. Between 1999 and 2005, the AAMR remained stable,

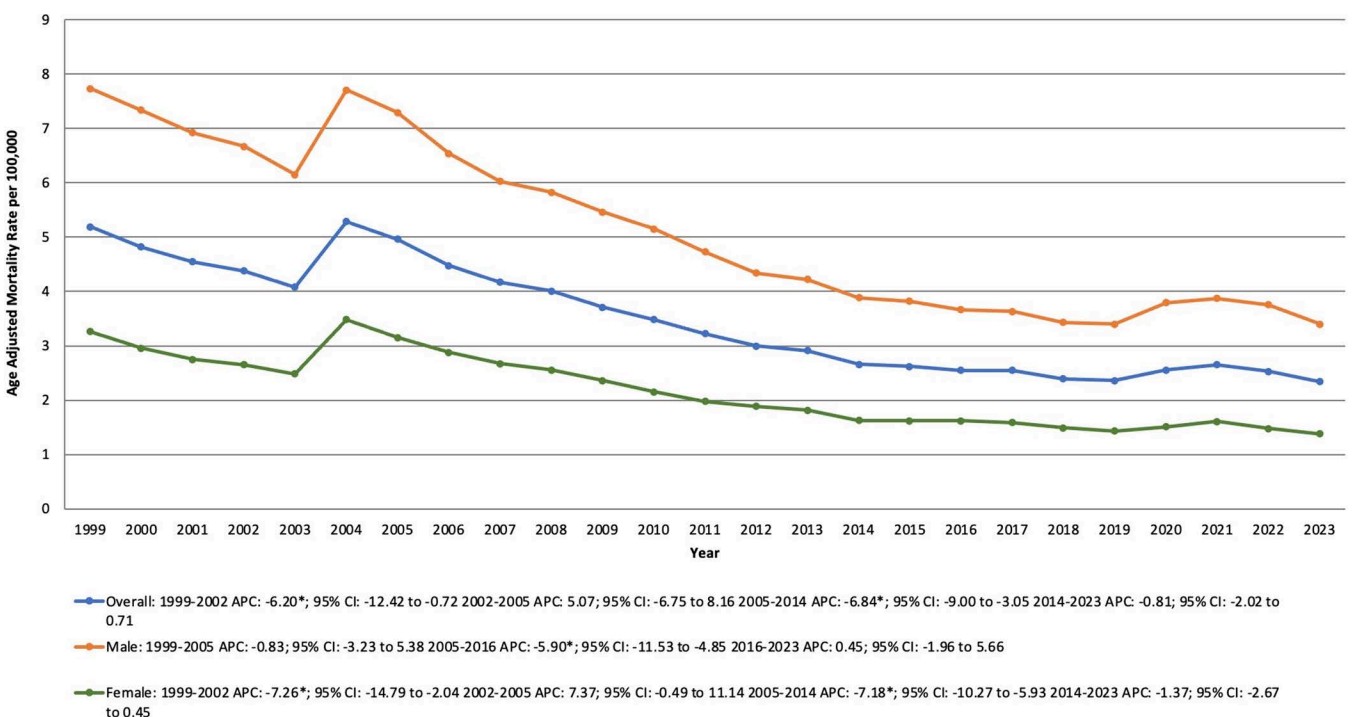

**Fig 1. Dilated Cardiomyopathy-related mortality trends from 1999 to 2023, stratified by the Overall population and Sex in the United States.** APC: Annual Percentage Change; CI: Confidence Interval.

followed by a notable decline from 7.29 in 2005 to 3.66 in 2016 (APC −5.90% [95% CI, −11.53 to −4.85; P=0.0032]). Between 2016 and 2023, the AAMR remained stable, with a slight increase from 3.66 to 3.4.

From 1999 to 2023, the AAMR among females decreased from 3.26 in 1999 to 1.38 in 2023. Between 1999 and 2002, the AAMR declined steeply from 3.26 to 2.65 (APC −7.26% [95% CI, −14.79 to −2.04; P=0.0216]). From 2002 to 2005, the AAMR remained stable, with a slight increase from 2.65 to 3.15. Between 2005 and 2014, the AAMR decreased significantly from 3.15 to 1.63 (APC −7.18% [95% CI, −10.27 to −5.93; P=0.0176]). From 2014 to 2023, the AAMR remained stable, decreasing slightly from 1.63 to 1.38. ( S2 Table, S3 Table, Fig 1).

### 3.3. DCM-related AAMR stratified Race/Ethnicity

Over the study period, the highest AAMR was observed in NH Black or African American populations, followed by the NH White group, Hispanic and Latinos and lastly, the NH Other populations.

For the NH Black or African American populations, the AAMR remained stable between 1999 till 2005, followed by a significant decrease from 9.49 to 3.99 between 2005 and 2015 (APC −7.91% [95% CI, −10.86 to −6.95; P<0.000001]). After 2015, the AAMR remained stable, reaching 3.77 in 2023.

For the NH White individuals, the AAMR remained relatively stable from 1999 to 2005, with a slight increase from 4.51 in 1999 to 4.59 in 2005. Between 2005 and 2014, the AAMR declined sharply from 4.59 to 2.55 (APC −6.31% [95% CI, −9.14 to −2.96; P=0.0328]). After 2014, the AAMR stabilized, reaching 2.3 in 2023.

For the Hispanic or Latino population, the AAMR significantly declined from 4.9 in 1999 to 1.72 in 2016 (APC −5.43% [95% CI, −8.91 to −4.49; P=0.0100]). After 2016, the AAMR remained stable, reaching 1.67 in 2023.

Lastly, for the NH other population, the AAMR significantly declined from 3.73 in 1999 to 1.23 in 2023 (APC −4.32% [95% CI, −4.98 to −3.63; P<0.000001]). (S2 Table, S4 Table, Fig 2).

### 3.4. DCM-related AAMR stratified by geographical region

**Statewide.** Throughout the study period, significant statewide variation in DCM-related mortality was observed. From 1999 to 2020, states falling within the top 90th percentile for mortality rates included Hawaii, Nevada, District of Columbia, Delaware and Washington, while those in the bottom 10th percentile were Kentucky, Massachusetts, Colorado, Connecticut, and Nebraska. In the subsequent period from 2021 to 2023, the states with the highest mortality rates were Hawaii, Washington, Delaware, South Carolina, and Utah, whereas Connecticut, Kentucky, New Hampshire, Arkansas, and Massachusetts ranked in the lowest 10th percentile (S5 Table).

**Census region.** From 1999 to 2023, the highest DCM-related mortality rates were observed in the Western region, followed by the Midwest, the South and lastly, the Northeastern region.

For the Western region, from 1999 to 2023, the AAMR declined significantly from 5.49 in 1999 to 4.69 in 2023 (APC −2.59% [95% CI, −3.01 to −2.16; P<0.0001]).

For the Midwest, the AAMR declined from 5.73 in 1999 to 4.94 in 2002, followed by stability until 2005. From 2005 to 2014, the AAMR significantly decreased from 5.59 to 2.88 (APC −6.87% [95% CI, −10.32 to −2.70; P=0.0372]). After 2014, the AAMR remained unchanged, reaching 2.42 in 2023.

For the Southern region, From 1999 to 2023, the AAMR declined from 5.27 in 1999 to 2.31 in 2023. Between 1999 and 2002, the AAMR showed a non-significant decline from 5.27 to 4.48, followed by a period of stability from 2002 to 2005, with the AAMR increasing slightly to 5.31. A significant decline occurred between 2005 and 2014, as the AAMR dropped from 5.31 to 2.41 (APC −8.79% [95% CI, −11.40 to −1.17; P=0.0440]). From 2014 to 2023, the AAMR remained relatively unchanged.

Lastly, for the Northeastern region, From 1999 to 2023, the AAMR declined from 4.07 in 1999 to 1.83 in 2023. Between 1999 and 2005, the AAMR remained relatively stable, followed by a significant decline till 2018, with the AAMR decreasing from 3.69 to 1.61 (APC −6.25% [95% CI, −13.26 to −5.22; P=0.0008]). However, from 2018 to 2023, the trend remained relatively stable, with a slight increase in AAMR from 1.61 to 1.83. (S6 Table, Fig 3).

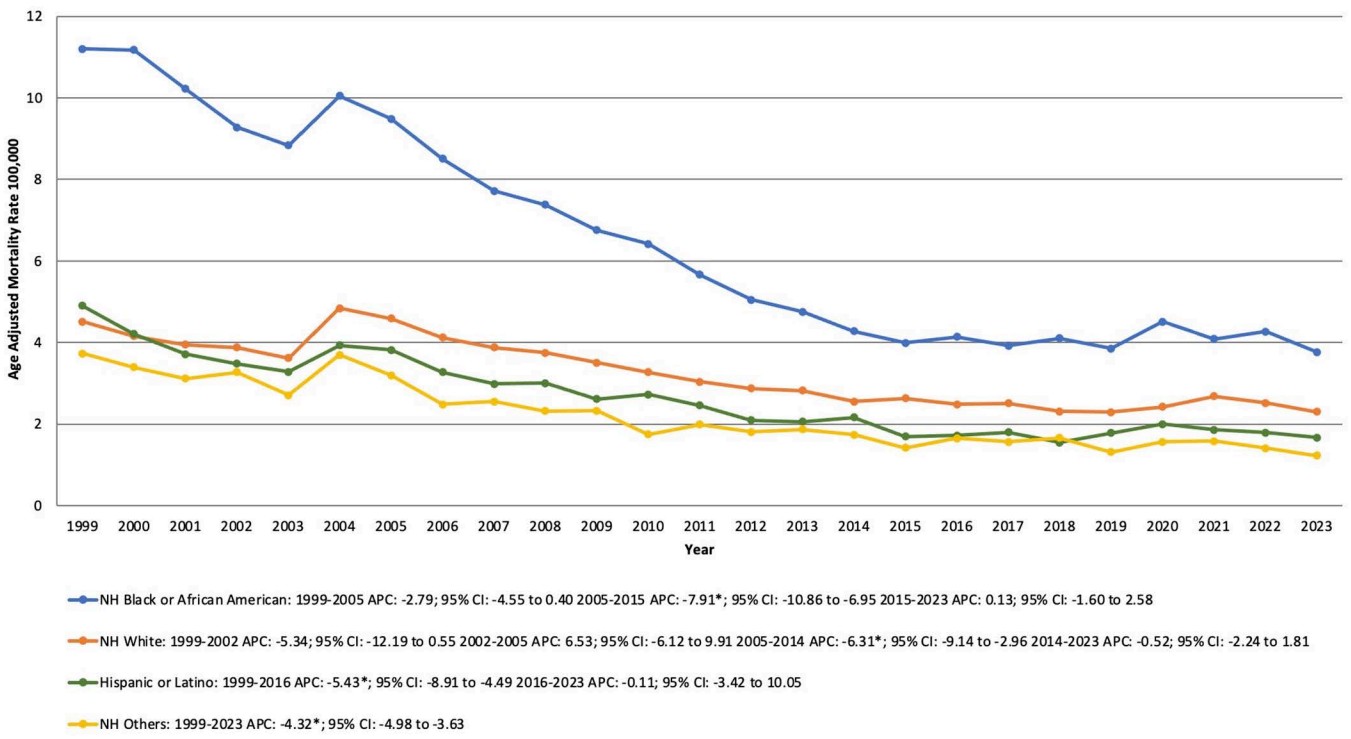

**Fig 2. Dilated Cardiomyopathy-related mortality trends from 1999 to 2023, stratified by the Racial groups in the United States.** APC: Annual Percentage Change; CI: Confidence Interval.

**Urban-rural.** Over the study period, rural areas averaged a considerably higher AAMR than urban areas. For rural areas, the AAMR showed a slight increase from 4.55 in 1999 to 5.19 in 2004 but remained relatively stable during this period. However, from 2004 to 2020, the AAMR significantly declined from 5.19 to 2.89 (APC −3.99% [95% CI, −6.83 to −3.20; P = 0.0024]). Similarly, for urban areas, the AAMR declined sharply from 5.31 in 1999 to 4.41 in 2002 (APC −6.55% [95% CI, −12.83 to −2.17; P = 0.0076]) followed by stability till 2005. Between 2005 and 2014, the AAMR experienced a significant decline from 4.96 to 2.64 (APC −6.93% [95% CI, −10.24 to −5.81; P = 0.0192]). After 2014, the rates remained relatively stable, reaching 2.53 in 2020. (S7 Table, Fig 4).

## 4. Discussion

This retrospective analysis of mortality data from the CDC WONDER database highlights several key findings. First, mortality rates related to DCM decreased from 1999 to 2014, followed by stabilizing through 2023. This trend was observed in both men and women. Second, NH Black and NH White adults exhibited the highest AAMRs for DCM-related deaths compared to other racial groups. Third, significant regional disparities were noted. From 1999 to 2020, states with the highest mortality rates (top 90th percentile) included Hawaii, Nevada, District of Columbia, Delaware and Washington, while those in the lowest 10th percentile included Kentucky, Massachusetts, Colorado, Connecticut, and Nebraska. In the subsequent period (2021–2023), the states with the highest mortality rates were Hawaii, Washington, Delaware, South Carolina, and Utah, whereas those with the lowest rates included Connecticut, Kentucky, New Hampshire, Arkansas, and Massachusetts. Additionally, mortality rates were consistently higher in rural areas compared to urban regions.

Our findings align with prior work by Ashraf et al. and Komminni et al., demonstrating a significant decline in DCM-related mortality from 1999 to 2014, followed by a period of stabilization [9,10]. However, our study extends the analysis

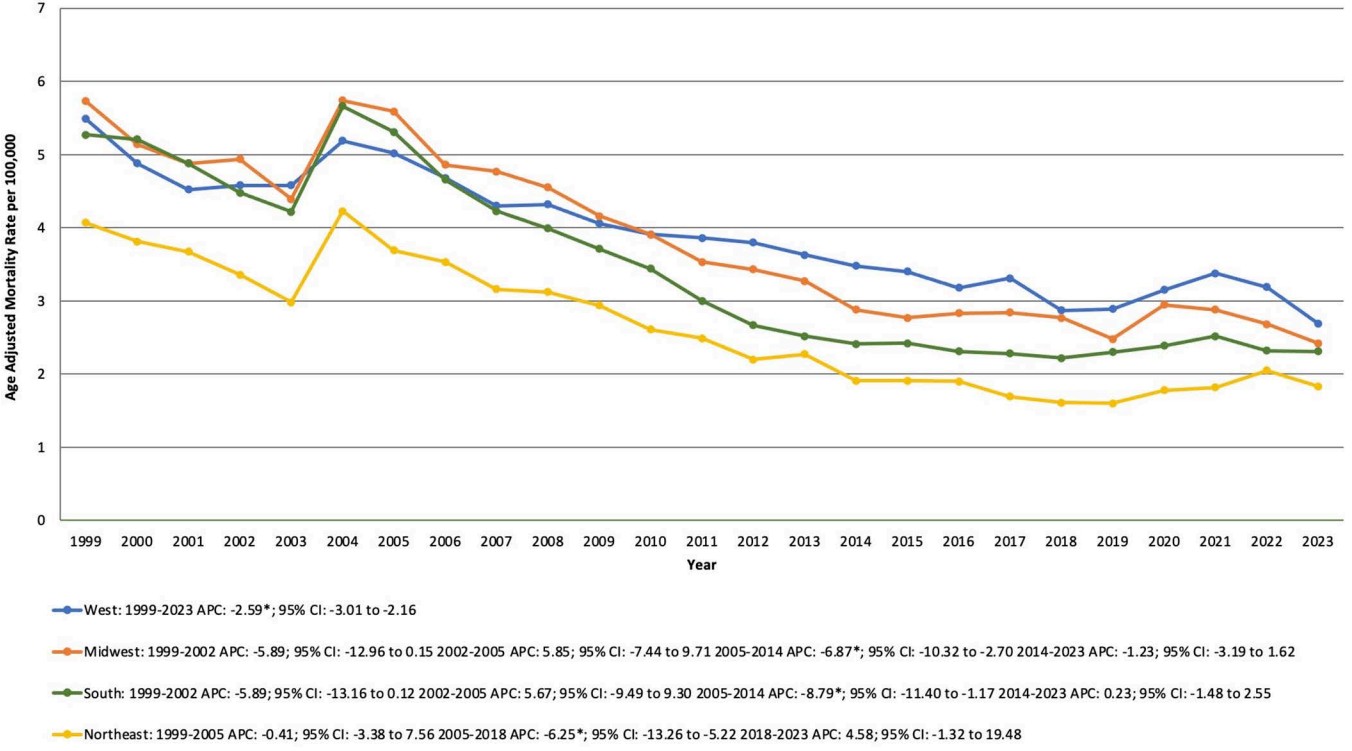

**Fig 3. Dilated Cardiomyopathy-related mortality trends from 1999 to 2023, stratified by Census Regions in the United States.** APC: Annual Percentage Change; CI: Confidence Interval.

through 2023; capturing the critical post-COVID-19 era; whereas previous studies were limited to trends up to 2020. Similar to Ashraf et al., we observed an overall decline from 2005 to 2014, followed by stabilization, though our longer follow-up period confirms that this plateau persisted through 2023. Sex-based analyses also showed similar patterns; however, unlike Ashraf et al., who reported a rise in female mortality between 2001 and 2004, our analysis found either declining or stable trends in women from 1999 to 2005 [10].

Both studies noted declining or stable mortality trends across racial groups, but differences emerged for the Hispanic population: Ashraf et al. described a continuous decline from 1999 to 2020, while we observed a steep decline from 1999 to 2016, followed by a plateau through 2023 [10]. Regional analysis further highlighted differences; our study found a consistent decline in AAMRs across all census regions from 1999 to 2023. In contrast, Ashraf et al. reported a significant increase in the Western region between 2001 and 2004 [10].

Our findings align with global trends in DCM-related mortality observed outside the United States. Zuin et al. conducted a retrospective analysis in Italy and reported an overall decline in DCM-related mortality from 2005 to 2017, with males consistently exhibiting higher age-adjusted mortality rates than females; mirroring our observations [17]. Similarly, Tsabedze et al. analyzed data from Sub-Saharan Africa and found that DCM was associated with a one-year all-cause mortality of 19%, with a disproportionate impact on males, consistent with the sex-related disparities identified in our study [18]. Furthermore, a retrospective study by Yu et al. in China analyzed outcomes among 6,453 patients with chronic heart failure over a mean follow-up of three years and reported that DCM had the highest mortality rate (44.9%) compared to other etiologies such as coronary artery disease (30.1%), hypertensive heart disease (36.2%), and rheumatic heart disease (13.1%) [19].

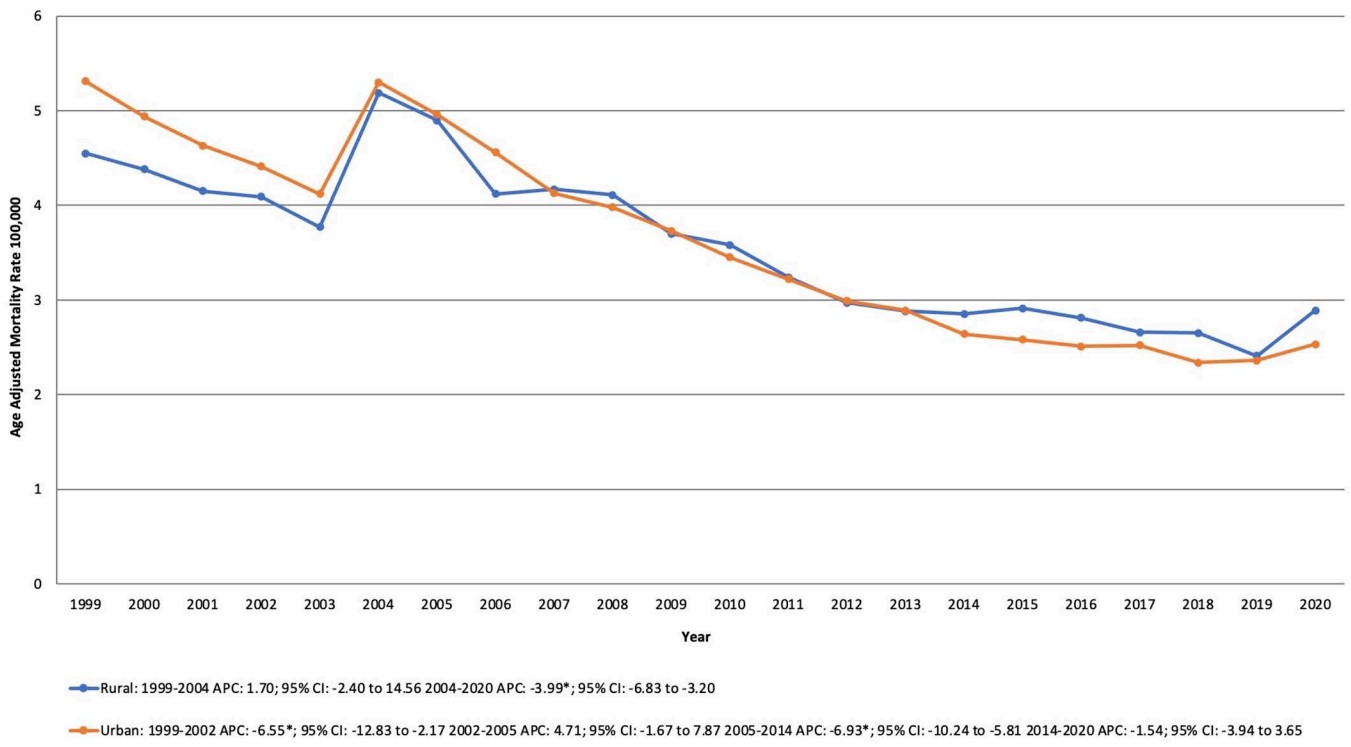

**Fig 4. Dilated Cardiomyopathy-related mortality trends from 1999 to 2023, stratified by Urbanization in the United States.** APC: Annual Percentage Change; CI: Confidence Interval.

With regard to other forms of cardiomyopathy, retrospective analyses have also been conducted. Renjitlal et al. examined mortality related to hypertrophic cardiomyopathy and reported trends similar to ours, noting an overall stable or declining pattern in the overall population [20]. Similarly, Raja et al. performed a mortality analysis focused on substance-induced cardiomyopathy and found a comparable declining trend in the overall population, mirroring the patterns observed in our study [21].

With regards to the impact of COVID-19 on cardiomyopathy, Baig et al. conducted a propensity-matched analysis to evaluate outcomes in patients with a history of non-ischemic cardiomyopathy (NICM). The study found no significant differences in in-hospital mortality, acute kidney injury (AKI), or pulmonary embolism (PE) between patients with and without NICM. However, those with NICM exhibited significantly higher rates of cardiovascular complications, including acute myocardial infarction, cardiogenic shock, cardiac arrest, mechanical ventilation, ventricular tachycardia, and ventricular fibrillation [22]. These findings suggest that while a prior diagnosis of NICM may not impact overall COVID-19 mortality, it is associated with an elevated risk of severe cardiovascular events during the course of infection.

Similarly, Iliuță et al. conducted a prospective study involving 142 patients with DCM, comparing clinical outcomes during the pre-pandemic period and the COVID-19 pandemic using a hybrid care model (in-person, online, and telemonitoring). The study reported no significant differences in blood pressure, body weight, symptom burden, adverse drug events, emergency room visits, or all-cause mortality between the two periods. However, patients with a restrictive left ventricular diastolic filling pattern (LVDFP) experienced significantly higher mortality and worse clinical status at two-year follow-up [23].

With regard to access to cardiovascular care during the COVID-19 era, Garcia et al. demonstrated that hospitalizations for acute myocardial infarction and cardiac catheterization laboratory activations for ST-segment elevation myocardial

infarction declined substantially during the pandemic [24]. This reduction was attributed to national stay-at-home orders, social distancing measures, reallocation of healthcare resources toward COVID-19 care, and patients' reluctance to seek medical attention due to fear of infection [24]. Similarly, Diamond et al. observed that patients receiving care during the initial COVID-19 surge were generally younger, had a higher burden of comorbidities, and experienced longer hospital stays compared to those seen in pre- and post-surge periods [25]. The first wave saw a significant decline in cardiac admissions; down to 29% of pre-surge levels; followed by a gradual return to baseline across all cardiac diagnoses [25]. Additionally, the widespread cancellation of outpatient cardiovascular visits likely delayed access to medication prescriptions, refills, and essential diagnostic testing [26,27]. Delays in semi-elective procedures such as transcatheter aortic valve replacement may have particularly impacted higher-risk cardiovascular patients. Furthermore, the rapid adaptation and resource reallocation in overwhelmed healthcare systems may have contributed to delayed or suboptimal inpatient and procedural care for non–COVID-19 patients [26,27]. Moreover, Uy-Evanado et al. indicated that delays in emergency medical service (EMS) response times and a reduction in bystander cardiopulmonary resuscitation (CPR) rates during the pandemic likely contributed to decreased survival following out-of-hospital cardiac arrests [28]. Lastly, the death certification practices also evolved during the pandemic. The introduction of the new ICD-10 code U07.1 for COVID-19 and CDC guidance led to shifts in how causes of death were recorded. This caused variability in attribution: cardiovascular deaths; including those due to cardiomyopathies; may have been under- or over-reported depending on whether COVID-19 was listed as the underlying or contributing cause [29,30].

This reduction in the overall DCM-related mortality can largely be attributed to advancements in DCM and HF treatments. A retrospective study by Castelli et al., spanning over three decades, demonstrated substantial improvements in survival among patients with idiopathic DCM [31]. In a cohort of 603 patients followed for an average of 8.8 years, the adoption of ACE inhibitors, β-blockers, and implantable devices increased significantly. As a result, mortality declined by 42% per treatment era, with the most recent cohort (2001–2011) experiencing an 86% reduction in HF-related deaths and an 87% reduction in sudden cardiac death compared to the earliest cohort (1977–1984) [31]. Similarly, a retrospective study by Merlo et al., which analyzed 853 patients across three treatment eras (1978–2007), reported a notable increase in the use of ACE inhibitors, β-blockers, ICDs, and CRT [32]. Over time, heart transplant-free survival at 8 years improved from 55% to 87%. Early diagnosis and personalized medical therapy were identified as key protective factors against HF-related mortality and the need for transplant, while ICD implantation significantly lowered the risk of sudden cardiac death [32]. Additionally, a retrospective study by Hagar et al., examining 1441 patients with IDCM from 2009 to 2016, observed a rise in the use of GDMT, including ACE inhibitors/ARBs, β-blockers, and aldosterone antagonists. Despite this progress, treatment remained suboptimal, with only 56% of patients receiving optimal therapy by 2016 [33]. Although GDMT was associated with better outcomes, there was no significant improvement in the overall long-term prognosis. However, factors such as higher LVEF, device therapy (ICD/CRT), and admission to a cardiology ward were linked to improved survival, reinforcing the necessity of comprehensive disease management [33]. These findings highlight the real-world benefits of evidence-based therapies and highlight the importance of long-term follow-up in optimizing patient outcomes, leading to an overall decrease in mortality rates associated with DCM.

Second, throughout the study period, males consistently exhibited significantly higher mortality rates than females. Male sex is a well-established risk factor for HF across various cardiovascular conditions, including DCM [34]. A comprehensive review by Jain et al., which analyzed sex-reported data from 31 studies on DCM, found a male-to-female ratio of 2.5:1 for non-genetic DCM and 1.7:1 for familial/genetic DCM across seven studies [35]. Similarly, the EuroHeart Failure Survey II reported a higher prevalence of DCM in males compared to females [36]. Likewise, a study by Halliday et al., which examined 881 patients with DCM (591 males and 290 females), identified a male-to-female ratio of 2:1, further emphasizing sex-based differences in DCM prevalence [37]. These disparities may be attributed to underlying sex-related pathophysiological mechanisms. Research has shown that men with DCM exhibit higher levels of apoptosis-related protein expression than women [38]. Additionally, cardiac MRI studies have revealed that men with myocarditis or acute DCM

have nearly twice the incidence of myocardial fibrosis compared to women, with fibrosis playing a critical role in DCM progression [39]. Sex differences in DCM have been largely linked to the influence of sex hormones on cardiac resident cells and inflammation. Experimental models lacking estrogen receptors suggest that estrogen has cardioprotective effects, enhancing reparative remodeling rather than promoting fibrosis—one of the key drivers of ventricular dilation and HF in patients with DCM [40,41].

In addition to hormonal influences, genetic factors have also been implicated in the higher incidence of DCM among males. A study by Haddad et al. observed sex differences in gene expression among patients with idiopathic DCM. In females, 55 genes—primarily involved in energy metabolism and transcription regulation—were differentially expressed (37 upregulated and 18 downregulated), whereas only 19 of these genes were expressed in males (13 upregulated and 7 downregulated). Notably, the dysregulated genes in males were associated with myocardial contraction [42]. While genes involved in ventricular remodeling were upregulated in both sexes, these sex-specific differences may serve as key indicators of disease progression and potential diagnostic markers or therapeutic targets for HF. Studies of familial DCM due to troponin T or titin mutations have shown a more severe phenotype in adult males than females, highlighting the role of genetic predisposition in sex-related differences [43]. According to Herman et al., In a cohort of 312 DCM patients and 249 controls, TTN mutations were identified in approximately 25% of familial and 18% of sporadic cases. While overall cardiac outcomes were similar between mutation carriers and non-carriers, adverse events occurred earlier in males, with a mean onset age of 56±3 years compared to 68±5 years in females [43]. These findings highlight the complex interplay between sex hormones, genetic factors and cardiac remodeling, which may contribute to the observed differences in DCM prevalence and outcomes between men and women.

Third, our analysis demonstrated that NH Black and NH White adults had the highest AAMR related to DCM throughout the study period, whereas NH Asians and American Indians exhibited the lowest rates. This disparity is consistent with existing literature highlighting a higher prevalence and incidence of DCM among Black individuals. A retrospective study by Coughlin et al. reported that African Americans are approximately 2.6 times more likely to develop idiopathic DCM compared to Whites (95% confidence interval [CI]: 1.6–4.3) [44]. Similarly, Towbin et al. found that the annual incidence of DCM in children under 18 years was 0.57 cases per 100,000, with a significantly higher rate among Black children compared to White children (0.98 vs. 0.46 cases per 100,000 per year), with most cases being idiopathic [45]. Additionally, Black individuals have a threefold increased risk of developing DCM and a twofold higher risk of mortality following diagnosis, independent of socioeconomic status and hypertension [5].The increased burden of DCM among African Americans is likely driven by both genetic and non-genetic factors. Hypertension and hypertensive cardiovascular disease—both more prevalent in African Americans—are well-established contributors to secondary DCM [46]. Similarly, diabetes, both diagnosed and undiagnosed, is more common among African Americans than Whites, with diabetic cardiomyopathy being a recognized cause of secondary DCM [46].

Recent genome-wide association studies have identified specific genetic variants that further increase DCM risk in Black individuals. One study estimated the heritability of DCM in Black patients to be approximately 33% and identified a novel intronic locus in the CACNB4 gene, which encodes a calcium-channel subunit essential for cardiac muscle contraction [47]. Additionally, four unique genetic variants in Bcl2-associated athanogene 3 (BAG3) were found almost exclusively in Black patients with DCM, and these variants were associated with a twofold increased risk of death or HF hospitalization [48]. Beyond biological factors, racial biases in healthcare and socioeconomic disparities further contribute to the disproportionate burden of DCM in Black individuals. Black patients are less likely to have private health insurance, attain higher education levels, or have a high household income compared to White or Hispanic patients [49]. In patients hospitalized for HF, Black individuals tend to be younger at admission, experience longer hospital stays, and have higher 90-day readmission rates than White or Hispanic individuals [50]. Furthermore, after orthotopic heart transplantation, Black patients face a 1.4-fold higher risk of graft failure and a 1.3-fold increased risk of mortality compared to White patients [51]. Importantly, Black patients are more likely to receive treatment at centers with higher-than-expected mortality rates, even after adjusting for insurance status and education level [50–52].

Fourth, with regards to geographical disparities, our analysis revealed an increased mortality rate in rural areas and the Western census region. Rural residents experience higher morbidity and mortality, with a 40% greater prevalence of heart disease compared to their urban counterparts [53,54]. Additionally, risk factors for DCM, including smoking, hypertension, diabetes, and obesity, are more prevalent in rural populations [53,54]. A retrospective study conducted by Manemann et al. across six Minnesota counties found that rural residence was independently associated with an increased risk of mortality among patients with HF (hazard ratio [HR], 1.18; 95% confidence interval [CI], 1.09–1.29) [55]. Similarly, a study by Turecamo et al. reported that rural participants had a 19% higher risk of developing heart failure compared to urban residents (HR, 1.19; 95% CI, 1.13–1.26) [56]. This elevated risk was particularly pronounced among women and Black men, persisting even after adjusting for cardiovascular risk factors and socioeconomic status [56]. This rural-urban divide in DCM related mortality may be largely attributed to socioeconomic disadvantages and disparities in healthcare access. Rural regions generally have lower socioeconomic status and face a significant shortage of healthcare professionals, particularly primary care providers and cardiologists, who are disproportionately concentrated in urban centers [57]. Between 2002 and 2015, the decline in primary care physicians in rural areas occurred at twice the rate observed in urban settings. Furthermore, the widespread closure of hospitals over the past decade—exacerbated by the COVID-19 pandemic—has further restricted access to essential cardiovascular care, deepening these existing healthcare inequities [57,58].

Lastly, our results indicated that the Western Census region had the highest mortality rates associated with DCM, potentially due to a higher disease burden. This trend may also reflect disparities in healthcare access, socioeconomic conditions, and the availability of specialized cardiac care in this region.

**Limitations and strengths.** This study has several notable limitations stemming from its reliance on the CDC WONDER database, which primarily uses death certificates to assess DCM-related mortality. The use of death certificates for nationwide surveillance may introduce inaccuracies, as some DCM-related deaths could be misclassified due to ICD coding discrepancies and the inherent limitations of relying solely on death certificate data. Additionally, the absence of comprehensive clinical information—such as LVEF disease severity, genetic predispositions, and specific treatment regimens—limits our ability to fully contextualize each case.

Furthermore, the study period spans the transition from ICD-9 to ICD-10 coding, which may have impacted the consistency and accuracy of cause-of-death reporting. Although this study did not evaluate the sensitivity, specificity, or predictive value of ICD codes, the shift in coding practices raises concerns regarding potential misclassification and ascertainment bias. The database also lacks crucial baseline characteristics of individuals, such as socioeconomic status, healthcare access, and racial disparities in treatment patterns, which may have influenced the observed mortality trends. Additionally, while AAMRs account for changes in population age structure, they may not fully adjust for evolving risk factors, advancements in diagnostic techniques, or shifts in heart failure management strategies that could impact DCM-related mortality. Despite these methodological limitations, the findings provide valuable insights into the burden of DCM-related mortality in U.S. adults and should be interpreted with caution.

However, this study also has several notable strengths. It is one of the first investigations to comprehensively evaluate long-term trends in DCM-related mortality in the United States, addressing a significant gap in the literature. Additionally, the study highlights disparities that may inform clinicians and healthcare policymakers, particularly regarding racial and geographic variations in mortality. The analysis also underscores the impact of healthcare resource distribution, with particular emphasis on rural and underserved regions. Moreover, the large dataset (N = 184,073) used to assess DCM-related mortality trends serves as a fundamental strength, offering a robust epidemiological perspective. Our analysis reveals a distinct temporal trend of declining DCM-related mortality from 1999 to 2014, followed by a plateau from 2014 to 2023. Further research is needed to explore the causal relationships between healthcare policies, socioeconomic factors, and these evolving mortality patterns to develop targeted interventions aimed at reducing DCM-related mortality further.

## 4. Conclusion

In conclusion, our retrospective analysis of CDC WONDER data reveals significant trends and disparities in DCM-related mortality over the past two decades. While overall mortality rates declined from 1999 to 2014 and stabilized thereafter, persistent sex-, race-, and geography-based disparities highlight ongoing challenges in disease management. Males exhibited consistently higher mortality rates, likely due to hormonal and genetic influences. NH Black and NH White adults had the highest AAMRs, reflecting both biological and socioeconomic determinants. Additionally, rural populations faced greater mortality burdens, underscoring healthcare access inequities. These findings emphasize the need for targeted interventions, equitable access to evidence-based treatments, and further research to address the underlying drivers of these disparities.

## Supporting information

**S1 Table. Dilated cardiomyopathy related deaths, stratified by sex and race in the United States, 1999–2023.**
(DOCX)

**S2 Table. Annual percent change (APC) of Dilated Cardiomyopathy age-adjusted mortality rates per 100,000 in the United States, 1999–2023.**
(DOCX)

**S3 Table. Overall and sex stratified dilated cardiomyopathy age-adjusted mortality rates per 100,000 in the United States, 1999–2023.**
(DOCX)

**S4 Table. Dilated cardiomyopathy related age-adjusted mortality rates per 100,000, stratified by race in the United States, 1999–2023.**
(DOCX)

**S5 Table. Dilated cardiomyopathy related age-adjusted mortality rates per 100,000, stratified by states in the United States, 1999–2023.**
(DOCX)

**S6 Table. Dilated cardiomyopathy related age-adjusted mortality rates per 100,000, stratified by census region in the United States, 1999–2023.**
(DOCX)

**S7 Table. Dilated cardiomyopathy related age-adjusted mortality rates per 100,000 in United States stratified by urban-rural classification, 1999–2020.**
(DOCX)

## Author contributions

**Conceptualization:** Mushood Ahmed, Syed Khurram M Gardezi.

**Data curation:** Ahmed Talaat Deiab, Aya Abouayana, Maryam Shahzad, Nabeel Ahmed.

**Formal analysis:** Zian Zafar Cheema, Ahmed Talaat Deiab, Aya Abouayana, Maryam Shahzad, Nabeel Ahmed.

**Investigation:** Mohammad Atout, Aya Abouayana, Maryam Shahzad.

**Methodology:** Hazim Mesmar, Asmaa Hasan, Maryam Shahzad.

**Project administration:** Taha Kassim Dohadwala, Maryam Shahzad, Mushood Ahmed.

**Resources:** Ahmed Talaat Deiab, Amaad Alam Shah.

**Software:** Nabeel Ahmed.

**Supervision:** Raheel Ahmed.

**Validation:** Zian Zafar Cheema, Raheel Ahmed.

**Visualization:** Raheel Ahmed.

**Writing – original draft:** Zian Zafar Cheema, Mohammad Atout, Taha Kassim Dohadwala, Ahmed Talaat Deiab, Hazim Mesmar, Asmaa Hasan, Muhammad Babar Mahmood, Daniel James Lewis, Hasan Ahmed.

**Writing – review & editing:** Maryam Shahzad, Mushood Ahmed, Raheel Ahmed, Syed Khurram M Gardezi.

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
