## [Decision Letter · Decision Letter 0]

7 Jul 2025

Dear Dr. Ahmed,

Thank you for submitting your manuscript to PLOS ONE. After careful consideration, we feel that it has merit but does not fully meet PLOS ONE’s publication criteria as it currently stands. Therefore, we invite you to submit a revised version of the manuscript that addresses the points raised during the review process.

Please address the minor concerns of one of the reviewers.

We look forward to receiving your revised manuscript.

Kind regards,

Aldrin V. Gomes, Ph.D.

Academic Editor

PLOS ONE

“None”

4. Please note that your Data Availability Statement is currently missing a direct link to access each database. If your manuscript is accepted for publication, you will be asked to provide these details on a very short timeline. We therefore suggest that you provide this information now, though we will not hold up the peer review process if you are unable.

Reviewers' comments:

Reviewer's Responses to Questions

**Comments to the Author**

1. Is the manuscript technically sound, and do the data support the conclusions?

Reviewer #1: Yes

Reviewer #2: Yes

2. Has the statistical analysis been performed appropriately and rigorously?

Reviewer #1: Yes

Reviewer #2: Yes

3. Have the authors made all data underlying the findings in their manuscript fully available?

Reviewer #1: Yes

Reviewer #2: Yes

4. Is the manuscript presented in an intelligible fashion and written in standard English?

Reviewer #1: Yes

Reviewer #2: Yes

Reviewer #1: General Assessment:

This manuscript presents a comprehensive population-based epidemiological analysis of dilated cardiomyopathy (DCM)-related mortality across the United States using CDC WONDER data from 1999 to 2023. The authors have assessed mortality trends stratified by sex, race/ethnicity, geography, and urban-rural classification, and used Joinpoint regression to calculate annual percent changes. The study is timely, well-executed, and presents critical findings that highlight disparities in DCM burden and mortality, particularly among Non-Hispanic Black individuals and rural populations.

The manuscript is overall scientifically sound and well-structured. However, contextualization with previously published work on this topic is essential to enhance the originality and relevance of the study.

Major Comments:

1. Comparison with Prior Literature on DCM Mortality (Incremental Value)

There are at least two peer-reviewed studies that have already examined trends in dilated cardiomyopathy-related mortality in the U.S. using the CDC WONDER database. These should be explicitly cited and discussed in both the Introduction and Discussion sections to clearly delineate how the current manuscript adds new insights:

• Ashraf, Taimoor MBBSa; Abunada, Omar MBBSb; Kumar, Aashish MBBSc; Ahmed, Saboor MBBSd; Ali Siddiqui, Muhammad Basit MBBSd; Memon, Umer MBBSe; Dev, Shah MBBSf; Meghjiani, Aashish MBBSd; Turesh, Muskan MBBSf; Khatri, Govinda MBBSd; Rai, Aneesh MBBSd; Manan, Abdul MBBSe; Deepak, Fnu MBBSg; Kumar, Mukesh MBBS, MPhilh; Yusuf, Salih Abdella MBBSi,*; Siddiq, Mohammad Arham MBBSe; Haseeb, Abdul MBBSe; Shafique, Muhammad Ashir MBBSe. Trends in mortality and disparities in dilated cardiomyopathy across gender, race, and region in the United States (1999–2020). Annals of Medicine & Surgery 2025 Feb;87(2):627–634. DOI: [10.1097/MS9.0000000000002908].

• Komminni, P; Raja, A; Raja, S; et al. TRENDS IN MORTALITY DUE TO DILATED CARDIOMYOPATHY IN THE UNITED STATES: A 1999–2020 CDC WONDER DATABASE STUDY. Journal of the American College of Cardiology (JACC). 2025 Apr;85(12_Supplement):1630. DOI: [10.1016/S0735-1097(25)02114-X].

These studies, while valuable, analyzed data only through 2020. The current manuscript provides three additional years (2021–2023), which is particularly important in the context of COVID-19-related disruptions in cardiovascular care. Additionally, this manuscript distinguishes itself by providing a more granular state-level and urbanization-based breakdown, and using Joinpoint regression with APCs for each demographic stratum—offering a more detailed and temporally segmented analysis.

The authors are strongly encouraged to include the above citations in their revised manuscript and clearly explain how the present analysis offers new and extended insights beyond these two prior works.

2. Validation of CDC WONDER for Cardiomyopathy Research

To support the reliability and credibility of the CDC WONDER database for cardiomyopathy mortality surveillance, it is advisable to cite studies that have used this dataset for related subtypes of cardiomyopathy. The following two references serve as examples of robust, peer-reviewed studies that utilized the same database to assess hypertrophic and substance-induced cardiomyopathy mortality:

• Renjihtlal SLM, Eid MM, Vyas C, Mohamed S, Shanmukhappa S, Renjith K, Mostafa MR, Baibhav B, Pillai N. Demographics and Trends of Hypertrophic Cardiomyopathy-Related Mortality in the United States, 1999–2020. Current Problems in Cardiology. 2023 Jul;48(7):101681. DOI: [10.1016/j.cpcardiol.2023.101681]. PMID: 36906160.

• Raja A, Raja S, Amin SB, Ahmed M, Rizvi SHA, Abdalla AS, Majid M, Asghar MS. Trends in substance-induced cardiomyopathy-related mortality among older adults in the United States from 1999 to 2020. Current Problems in Cardiology. 2024 Feb;49(2):102355. DOI: [10.1016/j.cpcardiol.2023.102355]. PMID: 38128635.

Incorporating these studies into the Methods or Discussion section will strengthen the argument that CDC WONDER is a validated and extensively used resource for studying cardiomyopathy epidemiology at the population level.

3. COVID-19 and Post-2020 Interpretation

Given the inclusion of data up to 2023, the authors should include a brief discussion of how the COVID-19 pandemic may have affected access to care, cardiovascular mortality, or death certification practices, potentially influencing the recent plateau or shifts in DCM mortality trends observed after 2020.

Reviewer #2: The article is good for publication except some mistakes like the binding words in « …considerably higherAAMR … »,

Furthermore, it would be great if the discussion included results from other continents/countries. E.g. results on the blacks compared with results from Africa, of Southern America…

**Do you want your identity to be public for this peer review?** For information about this choice, including consent withdrawal, please see our Privacy Policy

Reviewer #1: No

Reviewer #2: No

---

## [Author Response · Author response to Decision Letter 1]

15 Jul 2025

Response to Reviewers

Reviewer 1:

Comment 1: 1. Comparison with Prior Literature on DCM Mortality (Incremental Value)

There are at least two peer-reviewed studies that have already examined trends in dilated cardiomyopathy-related mortality in the U.S. using the CDC WONDER database. These should be explicitly cited and discussed in both the Introduction and Discussion sections to clearly delineate how the current manuscript adds new insights:

• Ashraf, Taimoor MBBSa; Abunada, Omar MBBSb; Kumar, Aashish MBBSc; Ahmed, Saboor MBBSd; Ali Siddiqui, Muhammad Basit MBBSd; Memon, Umer MBBSe; Dev, Shah MBBSf; Meghjiani, Aashish MBBSd; Turesh, Muskan MBBSf; Khatri, Govinda MBBSd; Rai, Aneesh MBBSd; Manan, Abdul MBBSe; Deepak, Fnu MBBSg; Kumar, Mukesh MBBS, MPhilh; Yusuf, Salih Abdella MBBSi,*; Siddiq, Mohammad Arham MBBSe; Haseeb, Abdul MBBSe; Shafique, Muhammad Ashir MBBSe. Trends in mortality and disparities in dilated cardiomyopathy across gender, race, and region in the United States (1999–2020). Annals of Medicine & Surgery 2025 Feb;87(2):627–634. DOI: [10.1097/MS9.0000000000002908].

• Komminni, P; Raja, A; Raja, S; et al. TRENDS IN MORTALITY DUE TO DILATED CARDIOMYOPATHY IN THE UNITED STATES: A 1999–2020 CDC WONDER DATABASE STUDY. Journal of the American College of Cardiology (JACC). 2025 Apr;85(12_Supplement):1630. DOI: [10.1016/S0735-1097(25)02114-X].

These studies, while valuable, analyzed data only through 2020. The current manuscript provides three additional years (2021–2023), which is particularly important in the context of COVID-19-related disruptions in cardiovascular care. Additionally, this manuscript distinguishes itself by providing a more granular state-level and urbanization-based breakdown, and using Joinpoint regression with APCs for each demographic stratum—offering a more detailed and temporally segmented analysis.

The authors are strongly encouraged to include the above citations in their revised manuscript and clearly explain how the present analysis offers new and extended insights beyond these two prior works.

Response 1: Thank you for your valuable feedback. We appreciate your recommendation to include prior peer-reviewed studies by Ashraf et al. and Komminni et al. that have examined DCM-related mortality trends using the CDC WONDER database. As suggested, we have now explicitly cited and discussed these studies in both the Introduction and Discussion sections of our revised manuscript.

Page 6-7; Lines 154 to 164 (Tracked Manuscript): “Although prior research has explored DCM-related mortality, data examining its demographic and regional distribution across the United States remain limited. Previous analyses using the CDC WONDER database, such as those by Komminni et al. and Ashraf et al., assessed trends in DCM-related mortality from 1999 to 2020 9,10. To build upon this work, we conducted an updated analysis extending through 2023, allowing for a comparison of trends before and after the COVID-19 pandemic. Identifying these patterns is crucial for addressing ongoing public health disparities. Therefore, we performed a retrospective nationwide study using CDC WONDER to assess DCM-related mortality trends among U.S. adults from 1999 to 2023, with a focus on variations by age, sex, race/ethnicity, and geographic region in order to identify high-risk populations and potential gaps in healthcare access and outcomes.”

Pages 13-14; Lines 312-326 (Tracked manuscript): “Our findings align with prior work by Ashraf et al. and Komminni et al., demonstrating a significant decline in DCM-related mortality from 1999 to 2014, followed by a period of stabilization 9,10. However, our study extends the analysis through 2023; capturing the critical post-COVID-19 era; whereas previous studies were limited to trends up to 2020. Similar to Ashraf et al., we observed an overall decline from 2005 to 2014, followed by stabilization, though our longer follow-up period confirms that this plateau persisted through 2023. Sex-based analyses also showed similar patterns; however, unlike Ashraf et al., who reported a rise in female mortality between 2001 and 2004, our analysis found either declining or stable trends in women from 1999 to 2005 10.

Both studies noted declining or stable mortality trends across racial groups, but differences emerged for the Hispanic population: Ashraf et al. described a continuous decline from 1999 to 2020, while we observed a steep decline from 1999 to 2016, followed by a plateau through 2023 10. Regional analysis further highlighted differences; our study found a consistent decline in AAMRs across all census regions from 1999 to 2023. In contrast, Ashraf et al. reported a significant increase in the Western region between 2001 and 2004 10”.

Comment 2: 2. Validation of CDC WONDER for Cardiomyopathy Research

To support the reliability and credibility of the CDC WONDER database for cardiomyopathy mortality surveillance, it is advisable to cite studies that have used this dataset for related subtypes of cardiomyopathy. The following two references serve as examples of robust, peer-reviewed studies that utilized the same database to assess hypertrophic and substance-induced cardiomyopathy mortality:

• Renjihtlal SLM, Eid MM, Vyas C, Mohamed S, Shanmukhappa S, Renjith K, Mostafa MR, Baibhav B, Pillai N. Demographics and Trends of Hypertrophic Cardiomyopathy-Related Mortality in the United States, 1999–2020. Current Problems in Cardiology. 2023 Jul;48(7):101681. DOI: [10.1016/j.cpcardiol.2023.101681]. PMID: 36906160.

• Raja A, Raja S, Amin SB, Ahmed M, Rizvi SHA, Abdalla AS, Majid M, Asghar MS. Trends in substance-induced cardiomyopathy-related mortality among older adults in the United States from 1999 to 2020. Current Problems in Cardiology. 2024 Feb;49(2):102355. DOI: [10.1016/j.cpcardiol.2023.102355]. PMID: 38128635.

Incorporating these studies into the Methods or Discussion section will strengthen the argument that CDC WONDER is a validated and extensively used resource for studying cardiomyopathy epidemiology at the population level.

Response 2: We thank the reviewer for highlighting the importance of validating the use of the CDC WONDER database in cardiomyopathy research. We have incorporated the suggested references (Renjihtlal et al., 2023 and Raja et al., 2024) into the Discussion section to support the credibility of CDC WONDER as a reliable data source for population-level analyses of cardiomyopathy-related mortality.

Page 15, lines 338 to 343 (Tracked manuscript): “With regard to other forms of cardiomyopathy, retrospective analyses have also been conducted. Renjitlal et al. examined mortality related to hypertrophic cardiomyopathy and reported trends similar to ours, noting an overall stable or declining pattern in the overall population 17. Similarly, Raja et al. performed a mortality analysis focused on substance-induced cardiomyopathy and found a comparable declining trend in the overall population, mirroring the patterns observed in our study 18”.

Comment 3: 3. COVID-19 and Post-2020 Interpretation

Given the inclusion of data up to 2023, the authors should include a brief discussion of how the COVID-19 pandemic may have affected access to care, cardiovascular mortality, or death certification practices, potentially influencing the recent plateau or shifts in DCM mortality trends observed after 2020.

Response 3: We thank the reviewer for this insightful suggestion. In response, we have added a paragraph to the Discussion section addressing how the COVID-19 pandemic may have influenced DCM mortality trends after 2020. Specifically, we discuss the potential impact of reduced healthcare access, delayed or deferred cardiovascular care, and evolving death certification practices during the pandemic.

Pages 15-17; Lines 344-383 (Tracked manuscript): “With regards to the impact of COVID-19 on cardiomyopathy, Baig et al. conducted a propensity-matched analysis to evaluate outcomes in patients with a history of non-ischemic cardiomyopathy (NICM). The study found no significant differences in in-hospital mortality, acute kidney injury (AKI), or pulmonary embolism (PE) between patients with and without NICM. However, those with NICM exhibited significantly higher rates of cardiovascular complications, including acute myocardial infarction, cardiogenic shock, cardiac arrest, mechanical ventilation, ventricular tachycardia, and ventricular fibrillation 17. These findings suggest that while a prior diagnosis of NICM may not impact overall COVID-19 mortality, it is associated with an elevated risk of severe cardiovascular events during the course of infection.

Similarly, Iliuță et al. conducted a prospective study involving 142 patients with DCM, comparing clinical outcomes during the pre-pandemic period and the COVID-19 pandemic using a hybrid care model (in-person, online, and telemonitoring). The study reported no significant differences in blood pressure, body weight, symptom burden, adverse drug events, emergency room visits, or all-cause mortality between the two periods. However, patients with a restrictive left ventricular diastolic filling pattern (LVDFP) experienced significantly higher mortality and worse clinical status at two-year follow-up 18.

With regards to access to cardiovascular care during the COVID-19 era, Garcia et al. demonstrated that hospitalizations for acute myocardial infarction and cardiac catheterization laboratory activations for ST-segment elevation myocardial infarction declined substantially during the pandemic 19. This reduction was attributed to national stay-at-home orders, social distancing measures, reallocation of healthcare resources toward COVID-19 care, and patients’ reluctance to seek medical attention due to fear of infection 19. Similarly, Diamond et al. observed that patients receiving care during the initial COVID-19 surge were generally younger, had a higher burden of comorbidities, and experienced longer hospital stays compared to those seen in pre- and post-surge periods 20. The first wave saw a significant decline in cardiac admissions; down to 29% of pre-surge levels; followed by a gradual return to baseline across all cardiac diagnoses 20. Additionally, the widespread cancellation of outpatient cardiovascular visits likely delayed access to medication prescriptions, refills, and essential diagnostic testing 21,,22. Delays in semi-elective procedures such as transcatheter aortic valve replacement may have particularly impacted higher-risk cardiovascular patients. Furthermore, the rapid adaptation and resource reallocation in overwhelmed healthcare systems may have contributed to delayed or suboptimal inpatient and procedural care for non–COVID-19 patients 21,22.

Moreover, Uy-Evanado et al. indicated that delays in emergency medical service (EMS) response times and a reduction in bystander cardiopulmonary resuscitation (CPR) rates during the pandemic likely contributed to decreased survival following out-of-hospital cardiac arrests 23. Lastly, the death certification practices also evolved during the pandemic. The introduction of the new ICD-10 code U07.1 for COVID-19 and CDC guidance led to shifts in how causes of death were recorded. This caused variability in attribution: cardiovascular deaths; including those due to cardiomyopathies; may have been under- or over-reported depending on whether COVID-19 was listed as the underlying or contributing cause 24,25”.

Reviewer 2:

Comment 1: The article is good for publication except some mistakes like the binding words in « …considerably higherAAMR … »,

Furthermore, it would be great if the discussion included results from other continents/countries. E.g. results on the blacks compared with results from Africa, of Southern America…

Response 1: We sincerely thank the reviewer for the valuable feedback. We have carefully revised the manuscript to correct typographical issues such as the binding error in “considerably higherAAMR. In response to the suggestion regarding broader contextualization, we have now expanded the Discussion section to include data from other regions outside the United States.

Page 9; Line 220 (Tracked manuscript): “males consistently had a considerably higher AAMR”.

Page 14-15, Lines 327-337 (Tracked manuscript): “Our findings align with global trends in DCM-related mortality observed outside the United States. Zuin et al. conducted a retrospective analysis in Italy and reported an overall decline in DCM-related mortality from 2005 to 2017, with males consistently exhibiting higher age-adjusted mortality rates than females; mirroring our observations 17. Similarly, Tsabedze et al. analyzed data from Sub-Saharan Africa and found that DCM was associated with a one-year all-cause mortality of 19%, with a disproportionate impact on males, consistent with the sex-related disparities identified in our study 18. Furthermore, a retrospective study by Yu et al. in China analyzed outcomes among 6,453 patients with chronic heart failure over a mean follow-up of three years and reported that DCM had the highest mortality rate (44.9%) compared to other etiologies such as coronary artery disease (30.1%), hypertensive heart disease (36.2%), and rheumatic heart disease (13.1%) 19.”

---

## [Decision Letter · Decision Letter 1]

16 Sep 2025

Trends and Disparities in Dilated Cardiomyopathy Related Mortality Among Adults in the United States: A CDC WONDER Analysis (1999-2023)

PONE-D-25-15148R1

Dear Dr. Ahmed,

We’re pleased to inform you that your manuscript has been judged scientifically suitable for publication and will be formally accepted for publication once it meets all outstanding technical requirements.

Kind regards,

Aldrin V. Gomes, Ph.D.

Academic Editor

PLOS ONE

Additional Editor Comments (optional):

Reviewer #2:

Reviewers' comments:

Reviewer's Responses to Questions

**Comments to the Author**

Reviewer #2: All comments have been addressed

2. Is the manuscript technically sound, and do the data support the conclusions?

Reviewer #2: Yes

3. Has the statistical analysis been performed appropriately and rigorously?

Reviewer #2: Yes

4. Have the authors made all data underlying the findings in their manuscript fully available?

Reviewer #2: Yes

5. Is the manuscript presented in an intelligible fashion and written in standard English?

Reviewer #2: Yes

Reviewer #2: Congratulations to authors !

The article has been refined and it is now good for publication in this journal.

**Do you want your identity to be public for this peer review?** For information about this choice, including consent withdrawal, please see our Privacy Policy

Reviewer #2: **Yes: ** Roland Muhindo Muyisa

---

## [Editor Report · Acceptance letter]

PONE-D-25-15148R1

PLOS ONE

Dear Dr. Ahmed,

I'm pleased to inform you that your manuscript has been deemed suitable for publication in PLOS ONE. Congratulations! Your manuscript is now being handed over to our production team.

Kind regards,

on behalf of

Dr. Aldrin V. Gomes

Academic Editor

PLOS ONE